# Immobilization of Phospholipase D on Silica-Coated Magnetic Nanoparticles for the Synthesis of Functional Phosphatidylserine

**Qingqing Han** [1,†], **Haiyang Zhang** [1,†], **Jianan Sun** [1,*], **Zhen Liu** [1], **Wen-can Huang** [1], **Changhu Xue** [1,2] and **Xiangzhao Mao** [1,2,*]

1  College of Food Science and Engineering, Ocean University of China, Qingdao 266003, China; qqinghan@126.com (Q.H.); brooks29@163.com (H.Z.); liuzhenyq@ouc.edu.cn (Z.L.); hwc@ouc.edu.cn (W.-c.H.); xuech@ouc.edu.cn (C.X.)

2  Laboratory for Marine Drugs and Bioproducts of Qingdao National Laboratory for Marine Science and Technology, Qingdao 266237, China

*  Correspondence: sunjianan@ouc.edu.cn (J.S.); xzhmao@ouc.edu.cn (X.M.);
   Tel.: +86-532-82031360 (J.S.); +86-532-82032660 (X.M.)

†  Qingqing Han and Haiyang Zhang contributed to the work equally and should be regarded as co-first authors.

**Abstract:** In this study, silica-coated magnetic nanoparticles ($Fe_3O_4/SiO_2$) were synthesized and applied in the immobilization of phospholipase D ($PLD_{a2}$) via physical adsorption and covalent attachment. The immobilized $PLD_{a2}$ was applied in the synthesis of functional phosphatidylserine (PS) through a transphophatidylation reaction. The synthesis process and characterizations of the carriers were examined by scanning electron microscope (SEM), transmission electron microscope (TEM), and Fourier-transform infrared spectroscopy (FT-IR). The optimum immobilization conditions were evaluated, and the thermal and pH stability of immobilized and free $PLD_{a2}$ were measured and compared. The tolerance to high temperature of immobilized $PLD_{a2}$ increased remarkably by 10°C. Furthermore, the catalytic activity of the immobilized $PLD_{a2}$ remained at 40% after eight recycles, which revealed that silica-coated magnetic nanoparticles have potential application for immobilization and catalytic reactions in a biphasic system.

**Keywords:** immobilization; phospholipase D; magnetic nanoparticles; stability; phosphatidylserine

## 1. Introduction

Phospholipase D (PLD, EC 3.1.4.4) is a lipolytic enzyme which can be used to hydrolyze phospholipids (PLs), and it can also catalyze the transphosphatidylation reaction, in the presence of an alcohol [1]. To date, many works were done to better understand PLD, mainly focusing on the purification and characterization of PLD from various sources, such as plants [2,3] and bacteria [4,5]. The analysis of PLD genes helps explain the relations between phospholipase domains and enzyme characterizations at the molecular level [6,7]. The heterologous expression of various PLD genes was reported in many kinds of microorganisms, mainly in *Escherichia coli* [8,9] and *Streptomyces* sp. [10]. The transphosphatidylation reaction is used to synthesize rare natural functional phospholipids, such as phosphatidylglycerol [11], docosahexaenoic acid-containing phosphatidylserine (DHA-PS) [12], phosphatidyl-glucose [13], and so on. Phosphatidylserine (PS), a functional phospholipid, is known to exert important physiological roles in humans [14]; it is therapeutically beneficial to improve brain function and can be used as an effective nutrient supplement in the food and pharmaceutical industries.

Meanwhile, as the main product of transphosphatidylation catalyzed by PLD, it attracted much focus on studying its transphosphatidylation process [15].

Although researchers made many efforts to make PLD more familiar to us, at present, the industrial application of the free enzyme is still limited due to instability and unrepeatability. With the increasing awareness of environmental protection, the application of enzyme immobilization technology is receiving more attention. Five immobilization techniques are mainly used, including covalent binding, adsorption, encapsulation, entrapment, and cross-linking [16–18]. As for the immobilization of PLD, some researchers reported binding PLD via adsorption to suitable supports, such as polyacrylamide gel, calcium gel, and macroporous resin [19–21]. PLD was also cross-linked to various carriers by glutaraldehyde [22,23]. Dittrich et al. immobilized PLD from *Streptomyces* sp. to aminopropyl-glass activated by glutaraldehyde and used it to produce phosphatidyglycerol [23]. Younus et al. [24] reported that a recombinant cabbage PLD was immobilized on cyanogen bromide (CNBr)-activated and antibody supports through covalent binding. However, it should be emphasized that there is no universal method for any particular enzyme in the industrial settings where simplicity and cost are required. Thus, it is necessary to search for and apply new materials and methods in the immobilization of PLD.

With the development of nanotechnology, nanomaterials became a new hot area in immobilization research, especially silica-coated magnetic nanoparticles (MNPs). Indeed, these materials have many important advantages such as superparamagnetism, low toxicity, large surface area, and easy separation from the reaction system [25]. In recent years, silica-coated magnetic nanoparticles were reported to immobilize dehydrogenase [26,27], protease [28,29], lipase [30–32], glucose oxidase [33], and other enzymes. The immobilization of enzyme on such magnetic solid materials often involves covalent coupling or non-specific adsorption techniques. Adsorption is mainly based on the hydrogen bonds and hydrophobic interactions, while covalent binding is based on the reaction between functional groups of the enzyme and the carriers, mainly via a carbodiimide linkage between enzymes and MNPs [18,27,34]. Yu et al. reported that $Fe_3O_4$-chitosan and $Fe_3O_4$-sodium alginate were used for the immobilization of $PLA_1$ [35] and $PLA_2$ [36], and the immobilized enzyme was applied to degum soybean oil.

Despite the fact that nanomaterials are widely used in enzyme immobilization, studies on the immobilization of PLD on such materials are surprisingly limited in the literature. In our previous work, we cloned and characterized $PLD_{a2}$ from *Acinetobacter radioresistens* a2 [37], whereby the conversion rate and selectivity of PS and DHA-PS were all about 100%. However, the biphasic system made it difficult to recover the free enzyme; thus, in this study, we immobilized the $PLD_{a2}$ on the surface of silica-coated magnetic nanoparticles to make better use of it. The optimum immobilization conditions and characterization of the immobilized $PLD_{a2}$ were investigated. Operational and storage stability of the immobilized $PLD_{a2}$ were evaluated as well. The results implied that the silica-coated magnetic nanoparticles could be used in the immobilization of $PLD_{a2}$; thus, the immobilization method described herein deserves further attention.

## 2. Results and Discussion

### 2.1. Characterization of Silica-Coated Magnetic Nanoparticles

The morphology and particle size of the silica-coated magnetic nanoparticles ($Fe_3O_4/SiO_2$) were observed by both scanning electron microscopy (SEM) and transmission electron microscopy (TEM). The SEM picture of $Fe_3O_4/SiO_2$ particles is shown in Figure 1A, illustrating that the size of $Fe_3O_4/SiO_2$ was smaller than 1 μm. After the reaction, the immobilized $PLD_{a2}$ was separated from the reaction system using a magnet (Figure 1B).

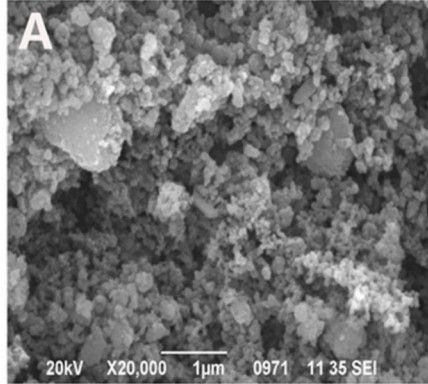
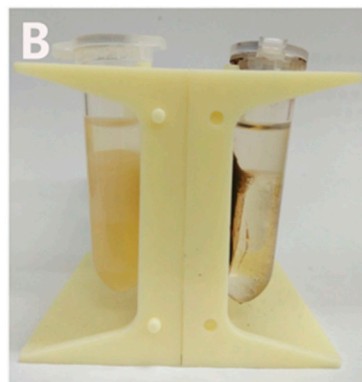

**Figure 1.** (**A**) Scanning electron microscopy picture of $Fe_3O_4/SiO_2$ particles. (**B**) Separation of immobilized $PLD_{a2}$ from reaction products using a magnet.

TEM pictures of the $Fe_3O_4$ and $Fe_3O_4/SiO_2$ particles are shown in Figure 2, the average sizes of $Fe_3O_4$ and $Fe_3O_4/SiO_2$ particles were about 10–15 nm and 50–60 nm, respectively. This might be explained by the magnetic properties of the particles, whereby they tended to form aggregates. After the coating process, the size of the nanoparticles increased. In Figure 2B, the black $Fe_3O_4$ particles seemed to be covered by a layer of a distinct gray material, which illustrated that the silica-coated magnetic nanoparticles were successfully created [30].

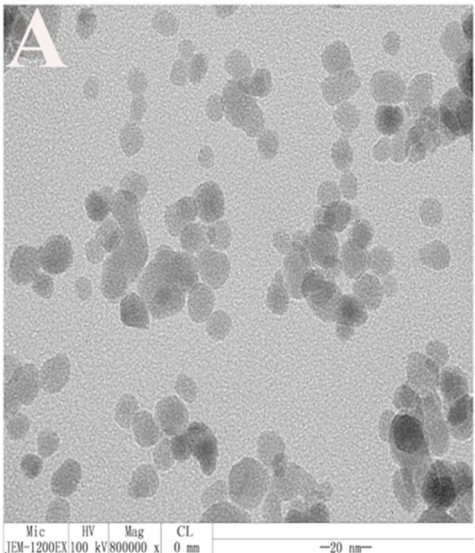
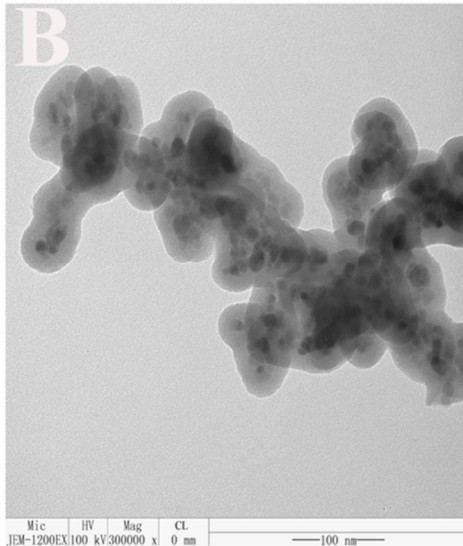

**Figure 2.** Transmission electron microscopy pictures of $Fe_3O_4$ (**A**) and $Fe_3O_4/SiO_2$ (**B**) particles.

Figure 3 shows the Fourier-transform infrared (FT-IR) spectra of the $Fe_3O_4/SiO_2$ particles (a), nanoparticles with bound $PLD_{a2}$ (b), and $PLD_{a2}$ (c), which confirmed the binding of $PLD_{a2}$ on the $Fe_3O_4/SiO_2$ particles, through the main band differences in the material and enzyme. There are five main characteristic peaks of pure PLD: (i) CONH peptide linkage (around 1650 cm$^{-1}$); (ii) CN stretching vibration of amines (around 1250 cm$^{-1}$); (iii) CH bonds (around 2950 cm$^{-1}$); (iv) COC groups (1100 cm$^{-1}$); (v) OH and NH vibrations (around 3300 cm$^{-1}$). As shown in Figure 3, there are strong bands at 1650 cm$^{-1}$ due to CONH peptide linkage ($PLD_{a2}$), as well as a broad peak around 1100 cm$^{-1}$ attributed to COC groups ($PLD_{a2}$) and SiO groups ($Fe_3O_4/SiO_2$). The results showed that $PLD_{a2}$ was successfully immobilized on the surface of silica-coated magnetic nanoparticles.

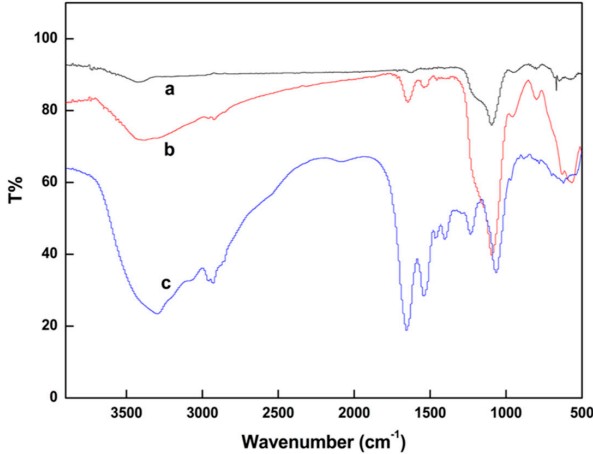

**Figure 3.** Fourier-transform infrared (FT-IR) spectra of $Fe_3O_4/SiO_2$ particles (a), $Fe_3O_4/SiO_2$ particles with bound $PLD_{a2}$ (b), and $PLD_{a2}$ (c).

### 2.2. Immobilized Conditions of $PLD_{a2}$

The optimum conditions for immobilization of $PLD_{a2}$ were investigated in three aspects (the initial $PLD_{a2}$ volume, the immobilization temperature, and the immobilization time), and the enzyme activity used for immobilization was 0. 25 IU/mg, while the enzyme concentration was 4.32 mg/mL. According to the immobilization of lipase on magnetic nanoparticles [30], 10 mg of $Fe_3O_4/SiO_2$ particles was firstly dispersed in different volumes of $PLD_{a2}$ liquid. As shown in Figure 4A, it was found that a combination of 1.0 mL of $PLD_{a2}$ liquid with 10 mg $Fe_3O_4/SiO_2$ particles obtained the highest enzyme activity. This might be due to the lower volume of $PLD_{a2}$, resulting in less protein absorbed to the carriers. However, the surface area of the carrier was limited, and too much enzyme liquid led to the coverage of the catalytic sites, thus, more enzyme did not equate to higher activity [18].

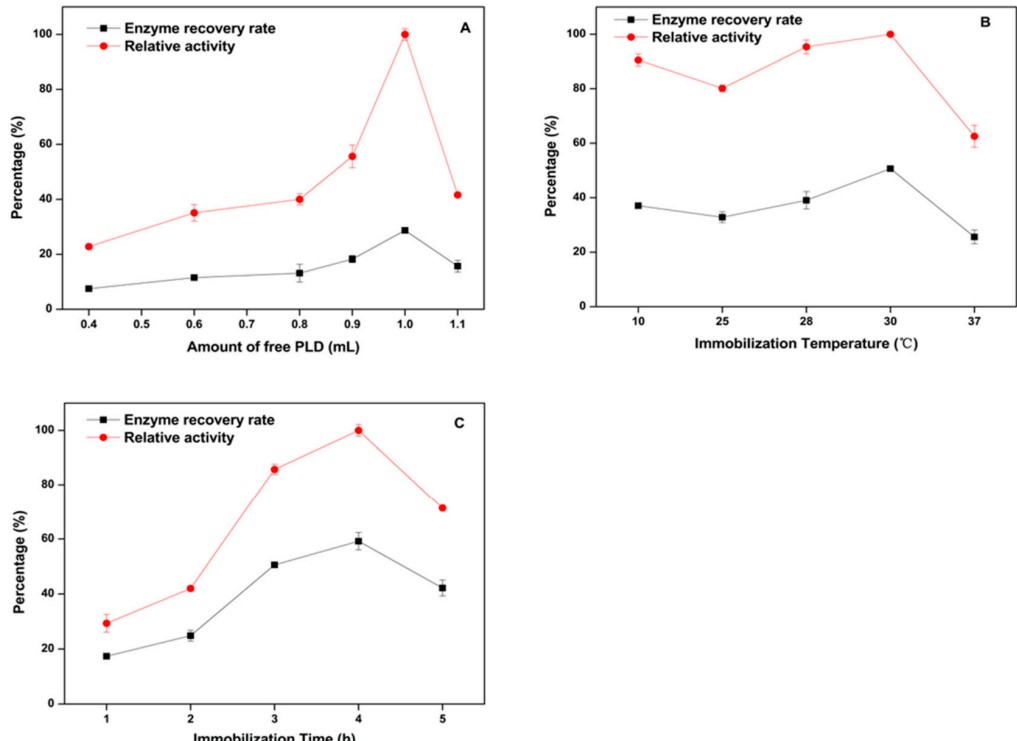

**Figure 4.** The effect of the initial $PLD_{a2}$ volume(A), temperature (B), and reaction time(C) on the enzyme recovery rate and the relative activity.

The optimum temperature of immobilization was also determined by the final enzyme recovery rate and the relative enzyme activity. About 10 mg of $Fe_3O_4/SiO_2$ particles was dispersed in 1.0 mL of $PLD_{a2}$ liquid and shaken at different temperatures for 5 h. In this part, the optimum temperature proved to be 30°C, as shown in Figure 4B.

As shown in Figure 4C, by increasing the reaction time from 1 to 5 h, the enzyme recovery rate increased, with reaction conditions of 1.0 mL of the initial $PLD_{a2}$ volume at a temperature of 30°C. The relative activity of immobilized PLD increased to maximum after 4 h. Compared to other methods, the immobilization time of this nanotechnology was shorter, as the covalent immobilization of PLD to VA-Epoxy Biosynth took 72 h [32] and it took 24 h to immobilize PLD on CNBr-activated supports [24].

Therefore, the immobilized $PLD_{a2}$ under the optimum conditions was further characterized and applied in a subsequent study. The enzyme recovery rate reached 59.16% at optimum conditions for the immobilization of $PLD_{a2}$. However, Ranjbakhsha et al. [30] reported the immobilization yield of lipase on such materials was 44.28% at optimum conditions.

### 2.3. Characterization of Immobilized $PLD_{a2}$

#### 2.3.1. Effect of Temperature on the Activity and Stability of Free and Immobilized $PLD_{a2}$

The optimum temperatures of the free and immobilized $PLD_{a2}$ are shown in Figure 5A. The optimum operational temperature of the immobilized $PLD_{a2}$ (50 °C) was raised by 10 °C compared to the free $PLD_{a2}$ (40°C) [37]. This might be explained by the protection effect of the carriers, which decrease the exposure of the immobilized $PLD_{a2}$ to temperature. As the temperature ranged from 20 to 60°C, the immobilized $PLD_{a2}$ showed less sensitivity to the change in temperature. This result is consistent with works described by Li et al. [38], where, after immobilizing PLD on non-porous nanoparticles, the optimum temperature (35°C) was raised by 5 °C compared to free PLD (30°C).

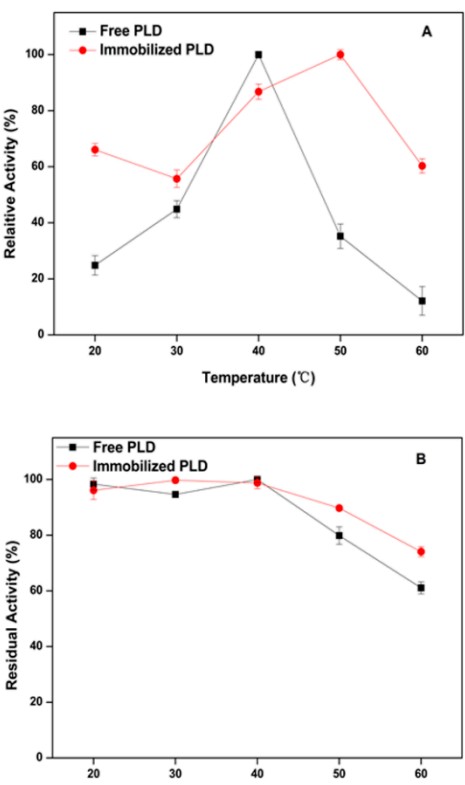

**Figure 5.** Effect of temperature on the enzyme activity (A) and stability (B) of the free and immobilized $PLD_{a2}$.

The thermal stability of the free and immobilized $PLD_{a2}$ in the range of 20–60°C is shown in Figure 5B, the results demonstrate that the immobilized $PLD_{a2}$ had better tolerance to high temperature and it was more stable than free $PLD_{a2}$ at different temperatures. However, immobilization does not ensure the improvement of enzyme characterizations. Younus et al. [24] showed that binding recombinant cabbage PLD to the antibody supports rendered the enzyme labile at high temperature. The better thermal stability of immobilized $PLD_{a2}$ may be due to the formation of multipoint interactions, since there are hydrogen, ionic, and hydrophobic interactions between the $PLD_{a2}$ and nanoparticles, which may protect $PLD_{a2}$ from deactivation [32].

### 2.3.2. Effect of pH on the Activity and Stability of Free and Immobilized $PLD_{a2}$

The results in Figure 6A reveal the optimum pH of free and immobilized $PLD_{a2}$, whereby both of them obtained the highest activity in acidic conditions. However, on the other hand, the immobilized $PLD_{a2}$ performed better in alkaline conditions. A similar phenomenon was reported by Lambrecht et al. [20], where it was shown that immobilization of PLD on octyl-sepharose resulted in an enlarged pH optimum range. This is possibly due to the support surface affording protection for the enzyme or the catalytic sites, rendering it less effective as pH changes in the reaction system.

The pH stability of the free and immobilized $PLD_{a2}$ was measured, and the results are shown in Figure 6B. It can be seen that, after being exposed to different pH for 12 h, the immobilized $PLD_{a2}$ was more stable and the enzyme activity changed less sharply.

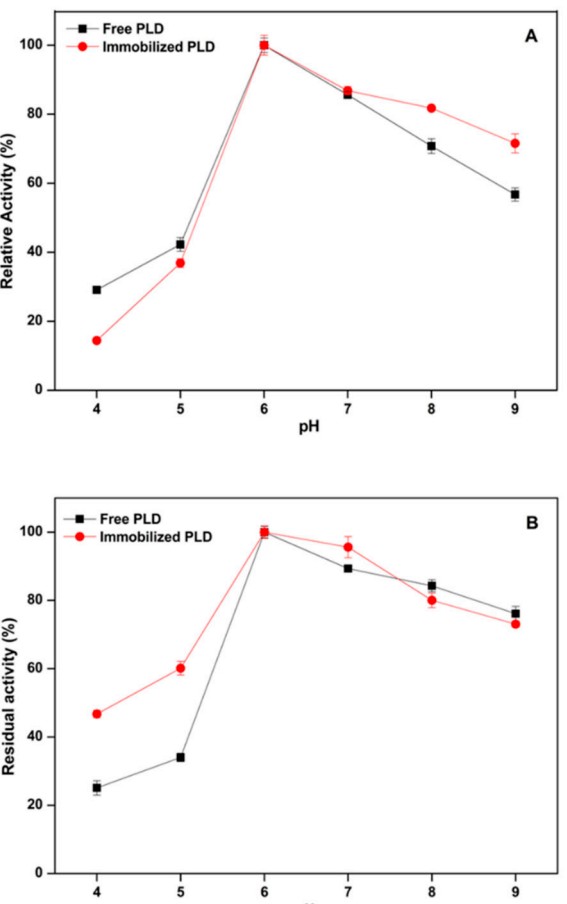

**Figure 6.** Effect of pH on the enzyme activity (A) and stability (B) of the free and immobilized $PLD_{a2}$.

### 2.3.3. Operational and Storage Stability of Immobilized PLD$_{a2}$

Reusability is a very important feature to evaluate the characteristics of immobilized PLD. There are two aspects to assess the operational stability of the immobilized PLD$_{a2}$: hydrolysis activity and the transphosphatidylation activity. At the end of each batch, the immobilized PLD$_{a2}$ was separated and recycled from the reaction system using a magnet. As shown in Figure 7, reusability presented a descending trend after seven times, whereby the hydrolysis activity remained higher than 40%, while the transphosphatidylation activity remained around 20%. The difference n results between these two measurement methods might be caused by the amount of immobilized PLD$_{a2}$ and the different reaction conditions. More enzyme was added to catalyze the transphosphatidylation reaction in a biphasic system, and damage of the enzymes was induced by organic solvents.

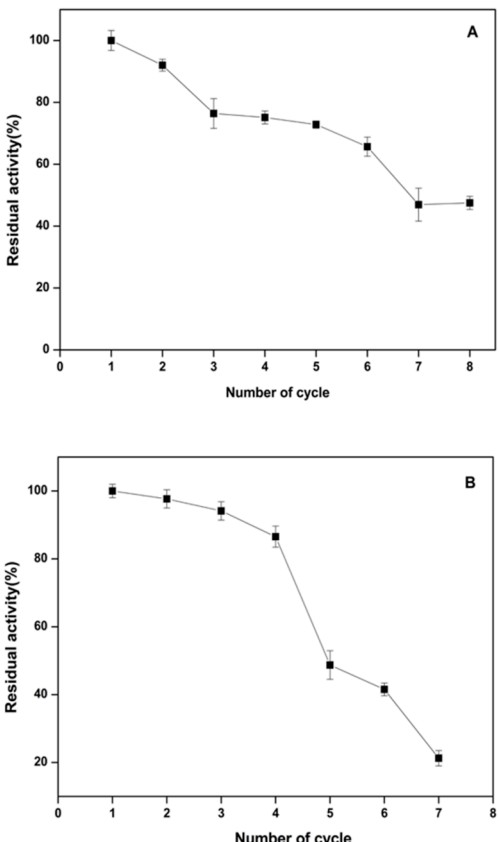

**Figure 7.** Operational stability of immobilized PLD$_{a2}$ in subsequent cycles of hydrolysis activity (**A**) and transphophatidylation (**B**).

Considering the production costs in industrial application, the reusability of an enzyme was proven to be a dominant characteristic of immobilization technology. The storage stability of immobilized PLD$_{a2}$ is shown in Figure 8, the immobilized PLD$_{a2}$ retained more than 60% of its initial activity after storage at 4°C for seven days. This indicated success in the immobilization of PLD on magnetic nanoparticles, in spite of a better effect on reusability and storage stability of the immobilized PLD$_{a2}$ in other reports [20,30].

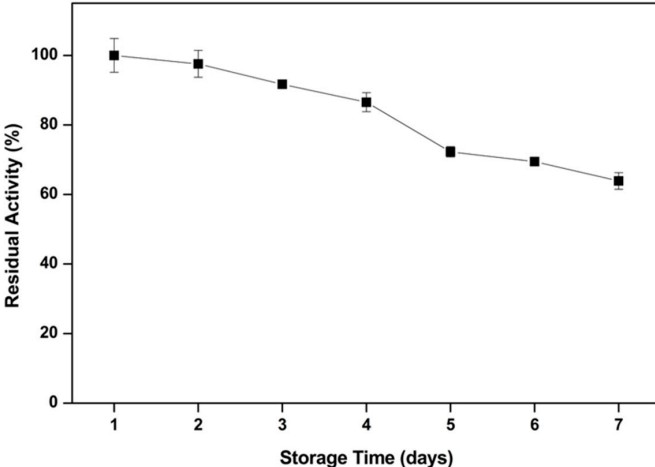

**Figure 8.** Storage stability of immobilized $PLD_{a2}$. The immobilized $PLD_{a2}$ was stored at 4°C for several days.

## 3. Materials and Methods

### 3.1. Materials

L-α-Phosphatidylcholine (95%) was purchased from Avanti (Alabama, USA); L-serine was purchased from Solarbio (Beijing, China); choline oxidase and peroxidase were purchased from Sigma-Aldrich (St. Louis, MO, USA). All other reagents used were of standard laboratory grade unless otherwise stated.

The strain *A. radioresistens* a2 was obtained from oil refineries, and the gene encoding for the $PLD_{a2}$ was cloned and expressed in *E. coli* BL21(DE3) [37].

### 3.2. Production of Enzyme PLD_{a2}

The strains were used as sources of recombinant *E. coli*, and $PLD_{a2}$ was cloned from *A. radioresistens* a2. The *E. coli* strains were grown in Luria–Bertani (LB) medium, which comprised (*w/v*) 0.5% tryptone, 0.1% yeast extract, and 0.1% NaCl, at 37°C with 50 μg/mL kanamycin (Solarbio, Beijing, China). Overexpression of $PLD_{a2}$ was induced by ZYP-5052 complete medium with kanamycin, incubating at 20°C for 48 h. The cells were sonicated in an ice bath and the $PLD_{a2}$ was collected by centrifugation. The enzyme samples for immobilization were prepared through filtering the crude extracts using a 0.45-μm ultrafilter.

### 3.3. Preparation and Characterization of Silica-Coated Magnetic Nanoparticles

The silica-coated magnetic nanoparticles were prepared according to the modified method described by Ranjbakhsh et al. [30]. Briefly, the co-precipitation method was used to prepare magnetic iron-oxide nanoparticles, and the nanoparticles were coated with silica via a sol–gel reaction. The size and structure of the silica-coated magnetic nanoparticles were determined by transmission electron microscopy (TEM) and scanning electron microscopy (SEM). Fourier-transform infrared spectroscopy (FT-IR) was used to ensure the $PLD_{a2}$ immobilized on the silica-coated magnetic nanoparticles.

### 3.4. Enzyme Immobilization on Silica-Coated Magnetic Nanoparticles

Firstly, 10 mg of silica-coated magnetic nanoparticles was taken for each sample, which was then dispersed in the enzyme liquid with various volumes (0.4 mL, 0.6 mL, 0.8 mL, 0.9 mL, 1.0 mL, and 1.1 mL). The mixture was shaken at different temperatures (10°C, 25°C, 28°C, 30°C, and 37°C) for 1–5 h. The immobilized $PLD_{a2}$ was recovered via magnetic separation, and the resulting immobilized $PLD_{a2}$ was washed with Tris-HCl buffer (20 mM, pH = 7.4) at least three times. Then, the activity of the

immobilized PLDa2 was measured, and the optimum conditions were based on the enzyme activity of the immobilized $PLD_{a2}$.

### 3.5. Characterization of Free and Immobilized Enzymes

To evaluate the effect of temperature on free and immobilized $PLD_{a2}$, the reactions were conducted at different temperatures (20°C, 30°C, 40°C, 50°C, and 60°C) at pH 7.4, and the enzyme activity was measured with other conditions unchanged. To determine the temperature stability, free and immobilized $PLD_{a2}$ samples were pre-incubated at 20–60°C with pH 7.4 for 4 h; then, the residual activity was measured under standard assay conditions.

As for the effect of pH on free and immobilized $PLD_{a2}$, the procedure was similar to the temperature experiments above. However, the reactions were conducted in a pH range of 4.0–9.0. To determine the pH stability, free and immobilized $PLD_{a2}$ samples were pre-incubated at 4°C at different pH (4.0–9.0) for 12 h; then, the residual activity was measured under standard assay conditions.

### 3.6. Operating Stability Assay

The operating stability of the immobilized $PLD_{a2}$ was measured by quantifying its catalyst activity in consecutive cycles of repeated use. After each batch reaction, the immobilized $PLD_{a2}$ was recovered by magnetic separation; then, the resulting immobilized $PLD_{a2}$ was washed with Tris-HCl buffer (20 mM, pH = 7.4) at least three times before being used for the next batch reaction. Through adding fresh substrates, the immobilized $PLD_{a2}$ was reused for a number of cycles. Both the hydrolysis activity and the transphosphatidylation activity of the immobilized $PLD_{a2}$ were measured.

### 3.7. Storage Stability Assay

The storage stability of the immobilized $PLD_{a2}$ was determined by its residual activity after incubation at 4 °C in Tris-HCl buffer (20 mM, pH = 7.4). The residual activity after different lengths of storage (one day, two days, three days, four days, five days, six days, and seven days) was assayed in conditions as described above.

### 3.8. Determination of Hydrolysis Activity

The enzyme activity of $PLD_{a2}$ constituted two assays: hydrolysis activity and transphosphatidylation activity [4]. The hydrolysis activity was measured using the method developed by Imamura and Horiuti with modifications [39]. The reaction mixture was composed of 100 μL of 10 mg/mL PC, 10 μL of 0.1 M citric acid buffer (pH 6.0), 5 μL of 0.1M $CaCl_2$, and 100 μL of the enzyme solution. The reaction was conducted at 37°C for 25 min, and stopped by adding 20 μL of 50 mM ethylenediaminetetraacetic acid (EDTA) in 1 M Tris-HCl buffer (pH 8.0). The resulting mixture was mixed with 200 μL of a solution of 0.5 U of choline oxidase, 0.2 U of peroxidase, 0.2 mg of 4-aminoantipyrine, 0.1 mg of phenol, and 2 mg of TritonX-100 in 1 M Tris-Hcl buffer (pH 8.0). After a 3-h reaction at 37°C, with the catalysis of choline oxidase and peroxidase, the intermediate products finally transformed to quinoneimine dye, whose absorbancy could be measured using an enzyme-linked colorimeter at 500 nm.

### 3.9. High-Performance Liquid Chromatography (HPLC) Assay

The transphosphatidylation reaction was performed in a biphasic system with PC and serine as substrates. The reaction mixture was composed of 1.0 mL of 0.2 M sodium acetate/acetic acid buffer (pH 6.2, including 1.0 M serine and 0.1 M $CaCl_2$) and 1.0 mL of 20 mg/mL PC dissolved in ethyl ether. The reaction was conducted at 40°C for 12 h. The transphosphatidylation activity was determined from the PS conversion ratio, defined as $100 \times PS/ (PC + PA + PS)$. The composition of the production after reaction was analyzed using an evaporative light scattering detector (ELSD) HPLC (Waters, USA). All HPLC experiments were conducted as follows: after nitrogen drying,

the products obtained from the reaction mixture were dissolved in hexane/isopropanol (81.42:17, *v/v*), then filtered using a 0.22-μm ultrafiltrate membrane. Parameters were as follows: 250 mm × 4.5 mm YMC DIOL column with 5-μm particle diameter, nitrogen as the nebulizing gas at a pressure of 25 psi, with a power level of 60% and a temperature setting of 50°C. The elution program was a nonlinear gradient with buffer A (hexane:isopropanol:acetic acid:triethylamine = 81.42:17:1.5:0.08 (*v/v/v/v*)) and buffer B (isopropanol:water:acetic acid:triethylamine = 84.42:14:1.5:0.08 (*v/v/v/v*)). The flow rate was 1 mL/min, the injection volume was 10 μL, and the column was equilibrated at 55°C. In these conditions, the proposed method was able to separate the reaction products as PA, PC, and PS.

## 4. Conclusions

Herein, we described the preparation method of silica-coated magnetic nanoparticles ($Fe_3O_4/SiO_2$) and their effective application for the immobilization of phospholipase D via physical adsorption and covalent attachment. FT-IR spectra proved the immobilization of PLD on the magnetic supports. The enzyme recovery rate reached 59.16% at optimum conditions (10 mg of silica-coated magnetic nanoparticles dispersed in 1.0 mL of enzyme liquid at 30°C for 4 h) for immobilization of $PLD_{a2}$.

In addition, the immobilized $PLD_{a2}$ had better tolerance to high temperature, and its optimum temperature was 10°C higher than that of free enzyme. Furthermore, the successful immobilization of PLD allowed its recovery and reuse in the synthesis of functional phosphatidylserine (PS). After reuse for the seventh time, the hydrolysis activity remained above 40%, while the transphosphatidylation activity remained around 20%. Therefore, silica-coated magnetic nanoparticles ($Fe_3O_4/SiO_2$) may have a promising future as materials for various biocatalyst reactions, and such immobilization methods could be applied in more fields.

**Author Contributions:** Conceptualization, J.S. and X.M.; data curation, Q.H. and H.Z.; formal analysis, Q.H. and H.Z.; funding acquisition, J.S., C.X., and X.M.; investigation, W.-c.H.; methodology, Z.L.; project administration, J.S., C.X., and X.M.; resources, C.X.; supervision, Z.L., W.-c.H., and X.M.; writing—original draft, Q.H.; writing—review and editing, H.Z.

**Funding:** This work was supported by the National Natural Science Foundation of China (31501516), the Taishan Scholar Project of Shandong Province (NO. tsqn201812020), the Applied Basic Research Program of Qingdao (16-5-1-18-jch), the Major Special Science and Technology Projects in Shandong Province (2016YYSP016), and the Laboratory for Marine Drugs and Bioproducts of Qingdao National Laboratory for Marine Science and Technology (LMDBKF201705).

**Conflicts of Interest:** The authors declare no conflicts of interest.

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
