# Peer review of "Immobilization of Phospholipase D on Silica-Coated Magnetic Nanoparticles for the Synthesis of Functional Phosphatidylserine"

_catalysts, doi:10.3390/catal9040361_

Reviewer 1 Report

Presented publication concerns the immobilization of phospholipase D on the surface of silica coated magnetic nanoparticles and application of the immobilized enzyme in the biosynthesis of phosphatidylserine through transphophatidylation reaction.

The revised version of the manuscript (original catalysts-400327) has been significantly improved as compared to the previous one. More detailed explanations are provided as well as recent references have been added and commented in the study.

Nevertheless, there are still some parts of the manuscript that require further improvements. For this reason see the comments below.

Materials and Methods section still requires improvements. More detailed information (such as amount, concentration or activity of the PLDa2 or amount of the free and immobilized enzyme used for the activity characterization) should be provided in the main manuscript to allow repetition of the experiments.

Also measurements of the enzyme activity should be described in details in Materials and Methods section.

Provided explanation of the enzyme-support interactions is insufficient. The Authors simply describe type of interactions characteristic for each of technique mentioned. More detailed explanation is required. For instance type of the chemical groups involved in the creation of adsorption or covalent interactions should be provided.

Line 137. Instead of organic solvents it should be temperature, as in this section Authors present effect of temperature on enzyme activity and stability.

Editorial and language issues still should be improved in the manuscript. In addition, references which have been added to the revised version should be properly formatted.

Author Response

1.   Materials and Methods section still requires improvements. More detailed information (such as amount, concentration or activity of the PLDa2 or amount of the free and immobilized enzyme used for the activity characterization) should be provided in the main manuscript to allow repetition of the experiments.

Response: Thank you for your suggestion. Some necessary information has been added in materials and methods section. The data about enzyme concentration was added in Pag.4, Line.109-110.

2.Also measurements of the enzyme activity should be described in details in Materials and Methods section.

 Response: Thank you for our suggestion. The measurements of the enzyme activity have been described in details in materials and methods section.

3.Provided explanation of the enzyme-support interactions is insufficient. The Authors simply describe type of interactions characteristic for each of technique mentioned. More detailed explanation is required. For instance type of the chemical groups involved in the creation of adsorption or covalent interactions should be provided.

 Response: Thank you for our suggestion. The detailed explanation has been added in Pag.2, line 61-64 in the revised manuscript.

4.   Line 137. Instead of organic solvents it should be temperature, as in this section Authors present effect of temperature on enzyme activity and stability.

Response: Thank you for your suggestion, organic solvents has been changed by temperature in the revised manuscript Pag 5, Line 139.

5.   Editorial and language issues still should be improved in the manuscript. In addition, references which have been added to the revised version should be properly formatted.

Response: Thank you for the comments. The language and grammatical mistakes have been checked carefully and have made some corrections in the revised manuscript. References have been checked carefully and have made some corrections in the revised manuscript.

Reviewer 2 Report

Authors have included the recommendations in this new version. However, English language should be still improved. Some samples of these corrections are listed below:

Abstract: line 13: nanoparticles 'were' synthesized instead of 'was'

-            Line 48 and 49: please, use Italics for et al. and Streptomyces

-            Line 55: 'a' should be included before 2 'new'

-            Line 80: 'that' should be included after 'illustrated'

-            Line 94: Please, rewrite the phrase... Figure 3 shows the FT-IR spectra

-       Line 108. Enzyme activity is now reported (0.25 IU/mg) as well as the volume of the enzyme employed in the immobilization (1 mL). It should be also included the data about enzyme concentration to know exactly the amount of enzyme that has been immobilized.

-          Line 116 ‘were’ should be included after ‘particles’

-          Line 117, shaken instead shake

-          Line 117, please use ‘at’ instead ‘in’

-          Line 120, please rephrase the sentence

-          Line 134 and Line 142, please employ dot after Fig. 5A and Fig. 5B respectively. English should be revised in these paragraphs.

-          Please check English language in section 2.3.3.

-          In sections 3.8 and 3.9 it would be recommendable to detail the reaction conditions (Substrates concentrations, amount of biocatalyst, temperature, pH, reaction volume)

Author Response

1.Authors have included the recommendations in this new version. However, English language should be still improved. Some samples of these corrections are listed below:

-Abstract: line 13: nanoparticles 'were' synthesized instead of 'was'

-            Line 48 and 49: please, use Italics for et al. and Streptomyces

-            Line 55: 'a' should be included before 2 'new'

-            Line 80: 'that' should be included after 'illustrated'

-            Line 94: Please, rewrite the phrase... Figure 3 shows the FT-IR spectra

-         Line 108. Enzyme activity is now reported (0.25 IU/mg) as well as the volume of the enzyme employed in the immobilization (1 mL). It should be also included the data about enzyme concentration to know exactly the amount of enzyme that has been immobilized.

-       Line 116 ‘were’ should be included after ‘particles’

-          Line 117, shaken instead shake

-          Line 117, please use ‘at’ instead ‘in’

-          Line 120, please rephrase the sentence

-          Line 134 and Line 142, please employ dot after Fig. 5A and Fig. 5B respectively. English should be revised in these paragraphs.

Response: Thank you very much. The language and grammatical mistakes have been checked carefully and have made some corrections in the revised manuscript. The data about enzyme concentration was added in Pag.4, Line.109-110.

2. Please check English language in section 2.3.3.

Response: Thank you for the comments. The language has been checked carefully and have made some corrections in the revised manuscript. The section 2.3.3 has been rewritten in the revised manuscript. 

3. In sections 3.8 and 3.9 it would be recommendable to detail the reaction conditions (Substrates concentrations, amount of biocatalyst, temperature, pH, reaction volume)

Response: Thank you for the suggestion. The details of the reaction conditions have been added in the revised manuscript.

This manuscript is a resubmission of an earlier submission. The following is a list of the peer review reports and author responses from that submission.

Round  1

Reviewer 1 Report

This manuscript reported the immobilization of Phospholipase D (PLDa2) on the surface of silica coated magnetic nanoparticles (Fe3O4/SiO2) via physical adsorption and covalent attachment, and application of immobilized PLDa2 in the synthesis of functional phosphatidylserine (PS) through transphophatidylation reaction. The enzymatic preparations were been characterized by SEM, TEM and FT-IR. Good results were obtained in catalytic activity after 8 cycle and in storage stability after 7 days.

The results of this work are interesting, but some minor points should be taken into consideration prior to publication:

- In my opinion, in the title of the manuscript the term biosynthesis is misleading. The term  biosynthesis refers to a chemical synthesis that occurs inside a living  organism, generally thanks to the catalysis of an enzyme. I suggest to change biosynthesis with synthesis.

- Pag. 1, line 15: Change biosynthesis with synthesis

- Pag. 10, line 286: Change biosynthesis with synthesis

Response: Thank you for your comments, biosynthesis has been changed to synthesis in the revised manuscript title, Pag.1, line 15 and Pag.10, line 270.

- Pag. 4, line 108 Change particleswith with particles with

Response: Thanks, particleswith has been changed to particles with in the revised manuscript line Pag.4, line104.

- Check the reference based on journal guidelines (Ref 6, 25, 31, 32, 33, 36, 37)

Response: Thank you for the comments. References have been checked carefully and have made some corrections in the revised manuscript.

- In the experimental section, the characterization of immobilized PLDa2, in term of activity (IU/mg of the protein) is missing. Please add this information.

Response: Thank you for your suggestion,  the enzyme activity used for immobilization was 0. 25 IU/mg” has been added in the revised manuscript Pag.4, line 108-109.

- Check the space before the references in all the manuscript.

Response: Thank you for the comments. The space before the references have been checked carefully and have made some corrections in the revised manuscript.

Reviewer 2 Report

Presented publication concerns the immobilization of phospholipase D on the surface of silica coated magnetic nanoparticles and application of the immobilized enzyme in the biosynthesis of phosphatidylserine through transphophatidylation reaction. In the study, the optimum immobilization conditions were evaluated and the thermal and pH stability of immobilized and free enzymes were measured and compared.

In my opinion presented manuscript is nice written and presents results that might be interesting. However, improvements and explanations are required before publication might be considered for publication in Catalysts. See the comments presented below.

I encourage Authors to provide some more detailed information related to support selection in enzyme immobilization. For this reason see the following references:

Chemical Society Reviews, 42, 2013, 6534-6565

Advances in Colloids and Interface Science, 258, 2018, 1-20

Catalysts, 2018, 8, 92

Molecules, 19, 2014, 8995-9018

Adsorption, 20, 2014, 801-821

Response: Thank you for the comments. These references have been added in the revised manuscript and some more detailed information has been added in the introduction section.

Lines 93-95. Reference is required for this statement.

Response: Thanks, references have been added in Pag.5 line 130 (Ref 30).

Section 3.1. Based on SEM pictures Authors claimed that magnetite-silica particles are "rather smaller than 1 um", as based on the TEM images They claimed that they are about 50-60 nm in diameter. It should be explained that due to the magnetic properties of the particles they tend to form aggregates.

Response: Thank you for the comments. The explaination has been added in Pag.3, line 87-88.

Lines 101-103. Wavenumbers values should be provided for these peaks.

Response: Thanks. Wavenumbers values have been added in Pag.3, line 96-99.

Lines 112-114. The amount, concentration or activity of the PLDa2 should be provided.

Response: Thank you very much. The enzyme activity used for immobilization was 0. 25 IU/mg and the value has been added in the revised manuscript in the revised manuscript Pag.4, line 108-109.

Lines 116-118. Reference is required for this statement.

Response: Thanks. Reference has been added in Pag.4, line 114 (Ref 18).

Figure 7. A and B should be provided in Figure caption.

Response: Thanks. Figure caption has been added in Pag. 8, line 177-178.

The possible explanation for the increase of the optimal temperature for the immobilized enzyme should be presented in the manuscript.

Response: Thank you for the comments. The explanation has been added in the revised manuscript Pag.5, line 136-137.

Section 3.4. The amount, concentration or activity of the PLDa2 should be provided.

Response: Thank you for your suggestion. The amount of the enzyme used for immobilization was 0.4-1 mL, the enzyme activity used for immobilization was 0.25 IU/mg and the value has been added in the revised manuscript.

Section 3.5. The amount of the free and immobilized enzyme used for the activity characterization study should be presented.

Response: Thank you for your suggestion. The amount of the free enzyme used for immobilization was 1 mL, the enzyme activity used for immobilization was 0.25 IU/mg.

In the Conclusions Authors write that enzyme was immobilized by adsorption and covalent binding. However, in the manuscript there is a lack of discussion about the mechanism of enzyme attachment. It should be discussed in the manuscript and mechanism of the binding should be presented.

Response: Thank you for the comments. The discussion about the mechanism of the binding has been added in the revised manuscript Pag.2 line 60-63.

Editorial issues should be improved in the manuscript.

There are some language and grammatical mistakes that need to be revised.

Response: Thanks. The language and grammatical mistakes have been checked carefully and have made some corrections in the revised manuscript.

Reviewer 3 Report

Han and coworkers describe the immobilization of a phospholipase D enzyme on silica coated magnetic nanoparticles. Authors report the improvement of the enzyme stability and activity after this immobilization strategy. However, diferent aspects should be improved before publication. - Abstract should be reduced. It is not necessary to content so detailed information.

Response: Thanks. The abstract has been re-written in the revised manuscript.

- English lenguage should be considerably improved. For example: - Line 31, ‘and’ should be included after the comma - Line 33, please, rephrase this sentence. - Line 34, bacteria instead of bacterias - Line 36, Expression instead of express - Line 48, cause should be replaced by causes. - Line 48, ‘on’ should be deleted. - The first paragraph of page 2 should be considerably improved - Line 64…’immobilize of’: delete ‘of’ Many other mistakes appear along the text and many paragraphs should be rewritten.

Response: Thank you very much. The language and grammatical mistakes have been checked carefully and have made some corrections in the revised manuscript. The first paragraph of page 2 has been rewritten in the revised manuscript.

In some cases the language difficults the understanding of the scientific results. Some other scientific aspects  are missed. As an example, the protein content is not reported, so it is not clear the amount of immobilized enzyme. How is the enzyme activity measured? no IU are reported. Thus, different aspects should be considerably improved before considering its publication.

Response: Thank you for your suggestion. The enzyme activity was measured with the methods described in Section 3.8 and 3.9, the enzyme activity used for immobilization was 0.25 IU/mg and the value has been added in the revised manuscript in the revised manuscript Pag.4, line 108-109. In our manuscript, the relative enzyme activity was used to make comparations in different conditions and showed clear trend.